# Maize Productivity and Household Welfare Impacts of Mobile Money Usage in Tanzania

Happiness Kilombele [1,*], Shiferaw Feleke [1], Tahirou Abdoulaye [2], Steven Cole [1], Haruna Sekabira [3] and Victor Manyong [1]

1 Social Science and Agribusiness, International Institute of Tropical Agriculture (IITA), Dar es Salaam 34441, Tanzania
2 Social Science and Agribusiness, International Institute of Tropical Agriculture (IITA), Bamako 320, Mali
3 Social Science and Agribusiness, International Institute of Tropical Agriculture (IITA), Kigali 1269, Rwanda
* Correspondence: h.zacharia@cgiar.org

**Abstract:** This study examined the determinants and impacts of mobile money (MM) usage on maize productivity and poverty likelihood (i.e., the probability of a household falling below the international poverty line at USD 1.9 per capita per day) in the Mbeya Region, Tanzania. The analysis was conducted using the endogenous switching regression (ESR) model on data from a random sample of 1310 households selected from seven districts in the region. Results of the ESR estimation show that MM usage is strongly and positively associated with the education level of the household head, asset ownership, credit access, input access, and social networks. MM usage is also significantly associated with increased maize productivity and a reduced poverty likelihood. Farmers who chose to use MM services increased their maize productivity by about 124 kg/acre and reduced their poverty likelihood by nearly 25 percentage points, as measured by the progress out of poverty index. These findings call for a targeted approach to reaching and supporting MM usage among households with constrained access to formal financial services to increase maize productivity and reduce poverty likelihood.

**Keywords:** mobile money; maize productivity; poverty likelihood; Tanzania

## 1. Introduction

The United Nations considers financial inclusion a key enabler in achieving many Sustainable Development Goals (SDGs). Financial inclusion ensures that unbanked households and businesses can access affordable financial services that meet their needs (World Bank 2022). People lack access to financial services mainly because they live in remote rural areas with no financial institutions, telecom infrastructure, or power supplies. The lack of access to financial services can affect agricultural productivity and economic welfare effects (Motta and Farias 2022). Communities with access to financial services experience lower poverty rates (Kast et al. 2018) and foster reduced gender inequalities (Arnold and Gammage 2019).

Mobile money (MM) adoption has significant and positive effects on financial inclusion in developing countries (Bongomin et al. 2021). In 2017, nearly 43% of the adult population (+15 years and above) in sub-Saharan Africa (SSA) owned an account at a financial institution or MM service provider, up from 23% in 2011 (Demirgüç-Kunt et al. 2017). Despite gains in financial inclusion, a gender access gap to financial services prevails in lower-income countries (Eckhoff et al. 2019), with the gap being most significant among the poor (Demirgüç-Kunt et al. 2013).

Expanded access to financial services through MM adoption is more evident in countries that adopted it earlier, such as Kenya, Uganda, and Tanzania. In 2017, 82% of the adult population in Kenya, 59% in Uganda, and 47% in Tanzania owned an account at a

financial institution or with an MM service provider, up from 42%, 21%, and 17% in 2011, respectively (Demirgüç-Kunt et al. 2017).

MM provides an opportunity to save, spend, and transfer money through a short messaging service (SMS) (Van Hove and Dubus 2019). MM services include (i) cashing in at an MM agent, i.e., exchanging physical cash for e-money usable on a mobile phone; (ii) transferring e-money to another mobile phone number; (iii) paying for products or services using e-money; and (iv) cashing out, i.e., exchanging e-money for physical money at an MM agent (Batista and Vicente 2020).

The advent of MM has transformed the landscape of financial services by changing how people send/receive money, save, and manage risk (Fanta et al. 2016). MM makes financial services available, accessible, and affordable for the financially excluded segments of the population (Andrianaivo and Kpodar 2012). MM services are relatively inexpensive and reliable, potentially facilitating money liquidity and minimizing the risk of carrying cash (Economides and Jeziorski 2017). Farmers excluded from formal financial services (e.g., banks) can now use MM to save money, receive remittances from relatives or friends in urban areas, and transfer and receive payments for their products. Remittance (i.e., sending money by someone living in an urban area to family/friends living in a rural area) is made more accessible. This can be achieved via a simple text message. As such, MM addresses critical market failures related to access to traditional financial services, liquidity, and cash flow management (Chale and Mbamba 2014). The essential features of MM that have catalyzed its increased use are its ability to provide secure and convenient access to money at affordable rates (Ouma et al. 2017).

In Tanzania, MM services were first introduced in 2008, starting with M-Pesa (Vodacom), followed by Tigo-Pesa (Tigo), Airtel-Money (Airtel), Ezy-Pesa (Zantel), Halo-Pesa (Halotel), and T-Money (Tanzania Telecommunications Company Limited (TTCL)). MM services have been increasing rapidly in the volume and frequency of transactions, with adoption rates in urban areas reaching 65% and 25% in rural areas (FITS 2013). MM services available in Tanzania include saving, sending, receiving money, making payments for products and services, and withdrawing money instantly through text messages (Parlasca et al. 2022).

As the access to financial services expands significantly across a range of countries in and outside of Africa, a growing number of studies have assessed the effects of MM usage (e.g., Abiona and Koppensteiner 2020; Batista and Vicente 2020; Mahmoud 2019; Sekabira and Qaim 2017a; Suri 2017; Suri and Jack 2016; Munyegera and Matsumoto 2016; Murendo and Wollni 2016; Jack and Suri 2014; Kikulwe et al. 2014; Kirui et al. 2013; Adams and Cuecuecha 2013). These studies have investigated the link between livelihood improvements and mobile financial services and found a positive relationship. The plausible explanation for the positive link is that MM facilitates resource mobilization for productive investments (e.g., farm inputs) through different channels, such as savings, remittances, low transaction costs, risk sharing, and consumption smoothing. For example, Ouma et al. (2017) found that access to mobile financial services not only promotes the likelihood of saving at the household level but also significantly impacts the amounts saved. However, Duvendack and Mader (2019) found that the impacts of MM interventions in lower-income countries are small and variable. They indicated that although some services positively affect some people, MM may generally be no better than comparable alternatives.

Although MM services have been provided in rural Tanzania for over a decade, empirical evidence is lacking on their impacts on agricultural productivity and poverty reduction. Only a few studies have attempted to investigate MM usage in Tanzania (e.g., Abiona and Koppensteiner 2020; Mahmoud 2019; Economides and Jeziorski 2017; Tumaini 2016; Seetharam and Johnson 2015; Yao and Shanoyan 2018). To the best of our knowledge, no prior studies in Tanzania have examined the impacts of MM usage on crop productivity. The existing literature focuses on the productivity and welfare effects of agricultural technologies, not financial innovations. This study aims to fill this research gap and addresses three research questions: (i) what are the determinants of MM usage,

(ii) what are the impacts of MM usage on maize productivity gains, and (iii) what are the impacts of MM usage on poverty reduction likelihood? Specifically, this study tests three hypotheses: (i) a suite of demographic and socioeconomic factors influences MM usage; (ii) MM usage increases maize productivity; and (iii) MM usage reduces poverty likelihood among smallholder farmers. The hypotheses were tested using the endogenous switching regression (ESR) model, a more rigorous analytical approach that controls for unobserved heterogeneity and endogeneity in the covariates (Lee 1982). The ESR model was recently applied in the impact assessment of agricultural technologies (Tufa et al. 2021; Wossen et al. 2017; Manda et al. 2019; Tambo and Wünscher 2017).

The study data come from a sample of 1310 households randomly selected from a target population of members from 130 Village Community Banks (VICOBAs) using a two-stage sampling approach in which VICOBAs were first selected, followed by households. The VICOBAs were located in seven districts in the Mbeya Region, the top maize-growing region of Tanzania. The selection of the study region was also guided by the importance of MM usage in the agricultural sector, where 41% of the adult population meets most of their expenses through money generated from farming activities (FSDT 2017).

This study makes four contributions to advancing scientific knowledge on MM usage for improved livelihoods in rural Tanzania. First, it adds to the small but growing literature on the productivity and welfare effects of financial innovation in a rural setting. The results of such an assessment can provide important evidence to inform policy priorities to increase maize productivity and reduce poverty. This study contrasts with existing impact studies on agricultural technologies such as improved crop varieties and agronomic practices.

Second, unlike many previous studies, this study applied a more rigorous analytical approach that controls for unobserved heterogeneity in the covariates. Applying such an analytical model can generate unbiased parameter estimates for more reliable evidence-based policymaking.

Third, it sheds light on the differential effects of MM usage on two household types—male-headed and female-headed. The results can be used to target specific household types and tailor specific policy interventions.

Fourth, it has relied on a rich dataset collected from a representative sample of 1300 households in a significant maize-producing region of Tanzania, allowing us to examine the determinants of MM usage and productivity and the welfare effects of several policy-relevant factors beyond MM usage. These include access to credit, access to irrigation, access to a tractor, access to farm inputs, and social networks. The results of the analysis of these policy-relevant factors in the present study can complement the findings on MM usage, thus providing important evidence that informs the design of complementary financial and agricultural policy interventions.

The following section presents a theoretical framework and empirical strategy establishing the relationships between farmers' MM usage and maize productivity and poverty likelihood outcomes. Section 3 presents and discusses the empirical results, focusing on the average and heterogeneous effects of MM usage on maize productivity and poverty likelihood. The final section concludes and draws policy implications.

## 2. Methodology

### 2.1. Theoretical Framework

We modeled a farmer's decision to use MM services using the random utility theory, which states that individuals choose between two options based on the perceived utilities from each option. Farmers were assumed to compare the benefits of MM usage relative to the locally available financial services in deciding to use MM. These might include informal sources (e.g., local borrowing from friends and relatives) and formal sources, such as microfinance institutions and banks.

Following Greene (2003), we represent MM usage for the $i^{th}$ farmer as a dummy variable, $M$. Within this framework, it was assumed that a farmer will choose MM over alternative services if the expected utility or benefit from MM usage is higher than that

of other financial services. The MM option is considered quicker, safer, and cheaper than formal financial services, such as commercial banks, micro-finance institutions, and postal offices (IOM 2014; Economides and Jeziorski 2017). The reduced time and lower costs of accessing capital could increase the farmer's savings and investment in agricultural inputs (Kikulwe et al. 2014; Donovan 2012). Additionally, MM is associated with benefits such as secure savings, facilitating self-insurance, efficiently managing risks, and investing in agricultural inputs, which in turn can increase agricultural productivity and reduce poverty (Sekabira and Qaim 2017a; Murendo and Wollni 2016; Jack and Suri 2014). By facilitating money transfers and savings and lowering the cost of remittances, MM usage can improve savings for use in farm input acquisition, thus leading to increased maize productivity and a reduced poverty likelihood. MM usage can help overcome market access constraints and facilitate input, output, and labor market participation through increased use of purchased inputs, sales of outputs, and employment opportunities (Kikulwe et al. 2014). MM users are also more likely to save and use these savings to buy farm inputs than non-users (Schaner 2016).

Considering the benefits of the MM option vis-à-vis the alternatives, the net benefit from MM usage can be defined as $M_i^* = W_1 - W_0$, where $W_1$ represents the benefits associated with MM usage and $W_0$ represents the benefits associated with the alternatives. However, the net benefit $M_i^*$ cannot be directly observed. If the farmers perceive that the benefit of MM usage is more significant than that of the alternatives ($M_i^* > 0$), it is revealed by the farmers' observed usage of MM services ($M_i = 1$)). By contrast, if the farmers perceive that the benefit of MM usage is smaller than that of the alternatives ($M_i^* \leq 0$), it is revealed by the farmer's observed non-usage of MM services ($M_i = 0$). The latent variable, $M_i^*$, can be defined by observable characteristics $Z_i$ and unobserved characteristics $u_i$, as follows:

$$M_i^* = Z_i'\gamma + u_i; \; M_i = \begin{cases} 1 & \text{if } M_i^* > 0 \\ 0 & \text{otherwise} \end{cases} \tag{1}$$

where $M_i^*$ represents a latent continuous variable representing MM usage; $Z_i$ is a vector of covariates used to model MM usage; $\gamma$ represents a vector of parameters to be estimated; $u_i$ is the error term.

In this study, MM usage is defined by a farmer's ownership and use of an MM account with one of the six MM service providers (Vodacom, Tigo, Airtel, Zantel, Halotel, and Tanzania TTCL) and their use of the account in 2018. MM account ownership is contingent on having a registered SIM card from a service provider (e.g., Vodacom). Once customers have their SIM card registered by visiting an agent of a service provider, they can start using the available financial services, such as cash transfers. They can access their account by dialing feature codes.

Assuming a standard normal distribution for the error term, Equation (1) can be cast as a probit model (Equation (2)):

$$M_i = Z_i'\gamma + u_i \tag{2}$$

Assuming a linear relationship, the outcome equation can be cast as a function of a vector of exogenous variables and endogenous MM usage $M_i$ using Equation (3), given as:

$$Y_i = X_i'\beta + \vartheta M_i + \varepsilon_i \tag{3}$$

where $Y_i$ is an outcome variable (either maize productivity defined by maize yield in kg per acre or poverty likelihood defined by the progress out of poverty index (PPI)); $\beta$ represents a vector of parameters to be estimated; $M_i$ is defined as previously; $\vartheta$ is the coefficient associated with the MM usage; $\varepsilon_i$ is the error term.

Equation (3) assumes that MM usage is exogenously determined while potentially endogenous. Farmers' decision to use MM services is voluntary and based on individual self-selection. The self-selection bias makes it challenging to isolate the causal effect of MM usage on the outcome variables. MM users and non-users are not directly comparable because they may have systematically different characteristics. For example, MM users

may have well-to-do family members living in urban areas who can afford to send them remittances. As a result, MM users are less likely to be poor. The unobserved variable (e.g., wealth status of the family members of the MM users) embedded in the error term $\varepsilon_i$ may be correlated with $u_i$ $[Corr(u_i, e_i) \neq 0]$, in which case Equation (3) cannot be consistently and efficiently estimated using the Ordinary Least-Squares (OLS) estimator. A selection bias results from unobservable factors affecting both error terms in the selection equation ($u_i$) and the outcome equation ($e_i$). Since MM usage is potentially endogenous, the parameter estimates can be biased and inconsistent, leading to an erroneous conclusion about the impact of MM usage on the outcome variables.

Allowing for endogenous switching where the effect of MM usage involves differences in parameter estimates of the covariates, we can estimate the ESR model using either the two-step procedure (Maddala 1983) or the full information maximum likelihood (FIML) (Lee and Trost 1978). However, the latter is more efficient than the former (Lokshin and Sajaia 2004).

*2.2. ESR Model*

The ESR model consists of the MM usage equation (Equation (3)) and two linear equations (Equations (4a) and (4b)) for each outcome (maize productivity and poverty likelihood) in two regimes, given the MM selection criterion.

$$\text{Regime1} : Y_{1i} = \beta_1 X_{1i} + e_{1i} \text{ if } M_i = 1 \tag{4a}$$

$$\text{Regime 2} : Y_{2i} = \beta_2 X_{2i} + e_{2i} \text{ if } M_i = 0 \tag{4b}$$

where $Y_{1i}$ and $Y_{2i}$ are outcome variables for MM users and non-users, respectively. For example, in regime 1, we regress maize productivity against $X_i$ for MM users. In regime 2, we do the same for non-users. $X_i$ represents a vector of exogenous variables related to demographic, socioeconomic, and institutional household characteristics; $e_{1i}$ and $e_{2i}$ are random terms. The error terms of Equations (3), (4a), and (4b) are assumed to have a tri-variate normal distribution (Maddala 1983).

The ESR model addresses the issue of selection bias as a missing variable problem by including the inverse Mills ratio terms from the probit model (Equation (3)) estimation into the outcome equations (Equations (4a) and (4b)) (Heckman 1979).

The outcome equations with the Mills inverse ratio inserted in the equations (Equations (4a) and (4b)) can be given as

$$Y_{i1} = X_{i1}\beta_1 + \sigma_{e_1 u}\lambda_{i1} + \xi_{1i} \tag{5a}$$

$$Y_{i2} = X_{i2}\beta_2 + \sigma_{e_2 u}\lambda_{i2} + \xi_{2i} \tag{5b}$$

where $\lambda_{i1}$ and $\lambda_{i2}$ are Mill's inverse ratio, and $\xi_{1i}$ and $\xi_{2i}$ are random disturbances associated with outcomes for MM users and non-users. A statistically significant $e_1 u$ and $\sigma_{e_2 u}$ indicate the presence of sample selection bias (Maddala 1986).

Following Shiferaw et al. (2014), the expected values of the outcomes of the treated (MM users) under actual and counterfactual scenarios can be computed as Equation (6a) and Equation (6b), respectively:

$$E[Y_{i1}|M_i = 1] = X_{i1}\beta_1 + \sigma_{e_1 u}\lambda_{i1} \tag{6a}$$

$$E[Y_{i2}|M_i = 1] = X_{i1}\beta_2 + \sigma_{e_2 u}\lambda_{i1} \tag{6b}$$

The average treatment effect on the treated (ATT), which represents the impact of MM usage on maize productivity or poverty likelihood for MM users, is the difference between the expected values of the outcomes of MM users in actual (Equation (6a)) and counterfactual (Equation (6b)) scenarios, given as:

$$\text{ATT} = E[Y_{i1}|M_i = 1] - E[Y_{i2}|M_i = 1] = (\beta_1 - \beta_2)X_{i1} + (\sigma_{e_1 u} - \sigma_{e_2 u})\lambda_{i1} \tag{7}$$

In the context of this study, the ATT refers to the average effect of MM usage on the MM users in terms of maize productivity and poverty likelihood. As a first step to examine the impact of MM usage using the ATT, we conducted a Kolmogorov–Smirnov (K-S) test for first-order stochastic dominance to test the equality of the observed (Equation (6a)) and counterfactual (Equation (6b)) cumulative distributions.

### 2.3. Data Collection

Data for this study were collected from a sample of 1310 households randomly selected from 130 VICOBAs in the Mbeya Region in 2018. VICOBA is a community-based savings and loans scheme whereby members save together and provide mutual support and loans. As shown in Figure 1, the survey covered seven districts: Chunya, Ileje, Mbarali, Mbeya, Mbozi, Momba, and Rungwe. These districts comprised 15%, 17%, 11%, 24%, 8%, 16%, and 9% of the sample. A two-stage sampling procedure was applied to select the households, with the first stage involving the selection of VICOBAs and the second stage involving the selection of households. The survey solicited information from the household heads using a standard questionnaire coded in Computer Assisted Personal Interview (CAPI) software called Surveybe™. Data were then exported into Stata 14.0 (Stata Corp, College Station, TX, USA) software for analysis.

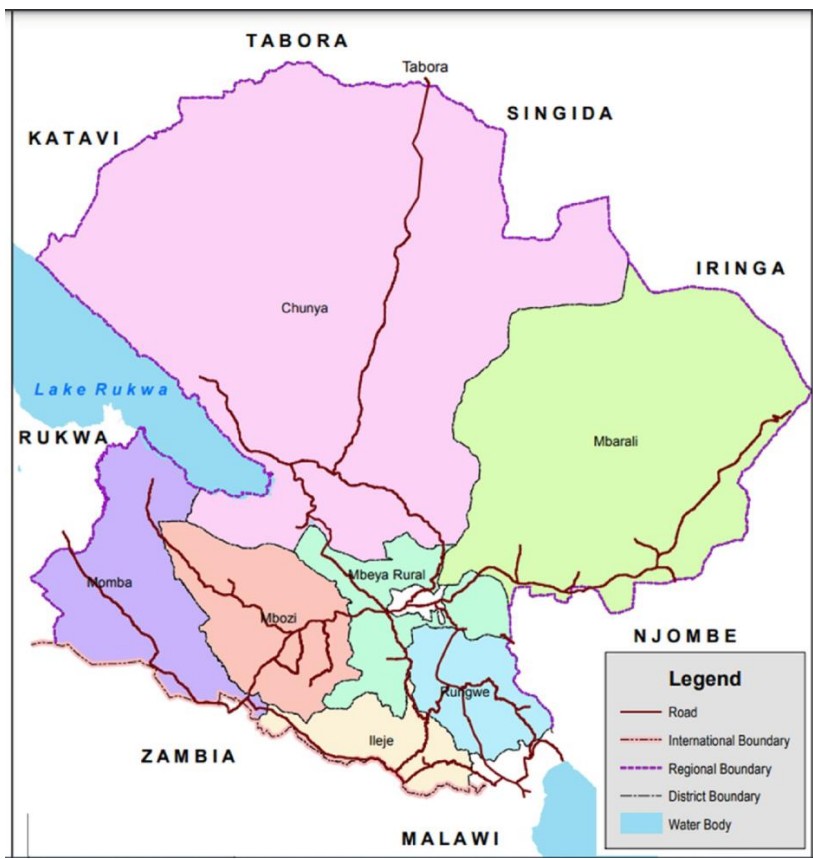

**Figure 1.** Map of the study region.

### 2.3.1. An Overview of VICOBA

The history of VICOBA in Africa dates back to the early 1990s when it was established in Niger as a women's group called "Mata Masu Dubara" (MMD), which means "Wise Women" (Lado Haule 2015). The formation of VICOBA was inspired by the success of the Cooperative for Assistance and Relief Everywhere's (CARE's) Village Savings and Loan Association (VSLA) program in Niger (Ahlén 2012). Many African countries have since adopted it to support women's agency and empowerment. In Tanzania, VICOBA was first introduced to Zanzibar in the early 2000s and has since expanded to different parts of

mainland Tanzania. There are about 24,123 community microfinance groups operating in Tanzania, of which 1260 are in the Mbeya region (Bank of Tanzania 2021).

VICOBA is an informal and community-based lending scheme with 25 to 30 self-selected members (SEDIT 2023). VICOBA members are individuals who have voluntarily come together to pool financial resources, acquire entrepreneurial skills, start new businesses, build a social network, and solve economic and social problems. The minimum age to become a VICOBA member is 18 years. Most members are less-educated, resource-constrained women with limited collateral to access loans from formal financial institutions (Bao et al. 2018). The benefits of becoming a VICOBA member include saving opportunities, access to soft loans for 3 to 6 months, access to business training opportunities, social solidarity, and women's agency and empowerment. Members acquire small business management skills and access concessional loans to invest in businesses or to meet living expenses such as health care costs and school fees. A VICOBA's financial resources come from members' contributions towards group savings. The management team raises funds from the sale of shares to members. Each member can purchase as many as three shares per week, which may be worth Tanzanian shillings 3000, 5000, 10,000, etc. (SEDIT 2023). Members play an active role in collaboratively setting loan terms and share values.

### 2.3.2. An Overview of Mbeya

Mbeya is one of the most productive maize-surplus-producing regions in the southern highlands of Tanzania, with an estimated productivity of 1.9 tons/ha (NBS 2021). Smallholder producers dominate the maize production sector, cultivating most of the total maize production in the region. Only a small number of large-scale commercial producers operate in the region (URT 2017). According to the latest agriculture census survey report (2019/2020), 293,993 tons of maize was produced in Mbeya, accounting for 4.5% of the total maize production in the country (NBS 2021). According to the 2022 Tanzania census report, the Mbeya region has a population of 2,707,410 people, and women head 36.3% of the households. The national average is 33.4 (or 33.5 in mainland Tanzania) (URT 2014). Agriculture is the mainstay of the region's economy, accounting for 40% of the regional economy and employing about 80% of the working population (URT 2017).

### 2.4. Variable Description and Measurement

The study had 1 treatment variable, 18 independent variables, and 2 outcome variables (Table 1). The treatment variable was MM usage, defined by farmers' use of MM services in 2018. The two outcome variables were maize productivity and poverty likelihood.

The maize productivity data were constructed from maize production data collected at the plot level. Respondents were first asked how many maize plots they managed in the year preceding the survey. In most cases, they had managed only one plot. Then, they were asked about the size of the maize plots (acres) and the level of outputs (in kg) obtained from the respective plots. Finally, the maize outputs from the different plots were aggregated into kg/acre for each household. The maize productivity data (yield in kg/acre) at the household level were converted into natural logarithms for maize productivity model estimation. The transformation into a logarithm reduces the skewness of the original productivity data and facilitates the convergence of maximum likelihood estimators.

Poverty likelihood was measured by the PPI developed by the Grameen Foundation (https://www.povertyindex.org/country/tanzania, accessed on 4 July 2022). The PPI measures poverty likelihood at the household level grounded on observable household characteristics, including assets and access to basic needs (Desiere et al. 2015). The PPI score was determined based on the values of answers to ten poverty-related questions in Tanzania's PPI scorecard (Table 2). The answers to the ten questions carry a specific weight based on a statistical model calibrated with data from the country's 2011/2012 household budget survey (see the weights in Table 2) (Schreiner 2016). Once the weights from all questions were summed up, a predefined table was used to look up the probability of a given household falling below the poverty line. In this study, we used the international

poverty line of USD 1.9 per capita per day adjusted for the International 2011 Purchasing Power Parity. The poverty likelihood is a non-negative integer that ranges from 0 (lowest likelihood of being poor at the international poverty line of USD 1.9 per capita per day) to 100 (highest likelihood).

**Table 1.** Descriptive statistics of variables by MM usage.

| Variable Description | Total | MM Users | Non-Users of MM | Mean Differences |
|---|---|---|---|---|
| Number of households | 1310 | 945 | 365 | |
| Natural log of maize productivity | 6.476 | 6.532 | 6.330 | 0.202 *** |
| | (0.796) | (0.783) | (0.813) | [0.050] |
| PPI score | 43.92 | 45.32 | 40.32 | 5.002 *** |
| | (11.05) | (10.38) | (11.89) | [0.708] |
| Age of household head | 44.02 | 43.19 | 46.19 | −3.000 *** |
| | (12.57) | (11.94) | (13.85) | [0.823] |
| Household type (1 = male-headed) | 0.889 | 0.906 | 0.844 | 0.062 *** |
| | (0.315) | (0.292) | (0.364) | [0.021] |
| Marital status (1 = married) | 0.921 | 0.939 | 0.877 | 0.062 *** |
| | (0.269) | (0.240) | (0.329) | [0.019] |
| Literacy level of household head (1 = able to read and write) | 0.892 | 0.935 | 0.778 | 0.157 *** |
| | (0.311) | (0.246) | (0.416) | [0.023] |
| Total household landholdings (acre) | 4.532 | 4.441 | 4.768 | −0.327 |
| | (4.457) | (4.347) | (4.727) | [0.285] |
| Tropical Livestock Units (TLU) | 1.173 | 1.229 | 1.030 | 0.199 * |
| | (2.174) | (2.351) | (1.625) | [0.114] |
| Television ownership (1 = yes) | 0.224 | 0.279 | 0.0795 | 0.200 *** |
| | (0.417) | (0.449) | (0.271) | [0.020] |
| Household size | 5.103 | 5.126 | 5.044 | 0.082 |
| | (1.594) | (1.575) | (1.642) | [0.100] |
| Dependency ratio (natural log of dep. ratio) | 4.473 | 4.468 | 4.487 | −0.019 |
| | (0.662) | (0.647) | (0.698) | [0.042] |
| Credit access (1 = yes) | 0.238 | 0.263 | 0.173 | 0.091 *** |
| | (0.426) | (0.441) | (0.378) | [0.024] |
| Tractor access (1 = yes) | 0.388 | 0.426 | 0.288 | 0.139 *** |
| | (0.487) | (0.495) | (0.453) | [0.029] |
| Access to irrigation water (1 = yes) | 0.221 | 0.223 | 0.216 | 0.007 |
| | (0.415) | (0.417) | (0.412) | [0.025] |
| Access to input dealer (1 = yes) | 0.362 | 0.344 | 0.408 | 0.001 |
| | (0.481) | (0.475) | (0.492) | [0.031] |
| Access to output buyers (1 = yes) | 0.497 | 0.497 | 0.496 | −0.064 ** |
| | (0.500) | (0.500) | (0.501) | [0.030] |
| Group membership (1 = yes) | 0.124 | 0.139 | 0.0877 | 0.051 *** |
| | (0.330) | (0.346) | (0.283) | [0.019] |
| Period of membership in VICOBA (years) | 0.376 | 0.363 | 0.408 | 0.368 *** |
| | (0.484) | (0.481) | (0.492) | [0.116] |
| Extension access (1 = yes) | 2.779 | 2.881 | 2.514 | −0.045 |
| | (2.017) | (2.089) | (1.790) | [0.030] |
| Wealth status (share proportion in VICOBA) | 0.129 | 0.137 | 0.107 | 0.030 *** |
| | (0.201) | (0.212) | (0.168) | [0.011] |
| Social networks in VICOBA (1 = know more than half of the members in VICOBA) | 0.787 | 0.817 | 0.710 | 0.107 *** |
| | (0.410) | (0.387) | (0.455) | [0.027] |
| Chunya District (1 = yes) | 0.160 | 0.179 | 0.112 | −0.093 *** |
| | (0.367) | (0.383) | (0.316) | [0.023] |
| Ileje District (1 = yes) | 0.136 | 0.110 | 0.203 | 0.058 *** |
| | (0.343) | (0.313) | (0.403) | [0.018] |
| Mbarali District (1 = yes) | 0.118 | 0.134 | 0.0767 | −0.018 |
| | (0.323) | (0.341) | (0.266) | [0.025] |
| Mbeya District (1 = yes) | 0.192 | 0.187 | 0.205 | 0.041 ** |
| | (0.394) | (0.390) | (0.405) | [0.020] |
| Mbozi District (1 = yes) | 0.137 | 0.148 | 0.107 | −0.090 *** |
| | (0.344) | (0.355) | (0.309) | [0.025] |
| Momba District (1 = yes) | 0.165 | 0.140 | 0.230 | 0.036 ** |
| | (0.371) | (0.347) | (0.421) | [0.016] |
| Rungwe District (1 = yes) | 0.0916 | 0.102 | 0.0658 | 0.04 ** |
| | (0.289) | (0.302) | (0.248) | [0.02] |

Standard deviation in parenthesis for column 2–4; standard errors in parenthesis for column 5. ***, ** and * denote significance at 1%, 5% and 10% respectively.

**Table 2.** Summary statistics for the PPI scorecard.

| Indicators | MM Users | MM Non-Users | Mean Differences |
|---|---|---|---|
| How many household members are 17 years old or younger? (0 = four or more, 10 = three, 15 = two, 20 = one, 30 = none) | 2.058 | 1.807 | 0.241 *** |
| | (1.393) | (1.470) | [0.089] |
| Do all children ages 6 to 17 years attend school? (0 = no, 3 = yes or no children in the household aged 6–17) | 10.69 | 10.76 | −0.098 |
| | (7.449) | (7.666) | [0.468] |
| Can the female head/spouse read and write? (6 = yes in Kiswahili, 13 = yes in English, 0 = no) | 5.899 | 4.983 | 0.944 *** |
| | (1.360) | (2.467) | [0.138] |
| What is the main building material of the floor of the main dwelling? (0 = earth, 11= concrete, cement, tiles, timber) | 7.944 | 5.956 | 2.020 *** |
| | (4.930) | (5.489) | [0.329] |
| What is the main building material of the roof of the main dwelling? (0= mud and grass, 8 = grass, leaves, bamboo, 9 = concrete, cement, galvanized corrugated iron sheets, asbestos sheets, tiles) | 8.840 | 8.624 | 0.268 *** |
| | (0.603) | (1.353) | [0.085] |
| How many bicycles, mopeds, motorcycles, tractors, or motor vehicles does your household own? (0 = none, 3 = one, 11 = two or more) | 0.707 | 0.481 | 0.223 *** |
| | (0.455) | (0.500) | [0.030] |
| Does your household own any radio or radio cassettes? (0 = no, 6 = yes) | 0.789 | 0.544 | 0.248 *** |
| | (0.408) | (0.499) | [0.029] |
| Does your household own any lanterns? (0 = no, 6 = yes) | 5.007 | 4.740 | 0.267 * |
| | (2.231) | (2.447) | [0.148] |
| Does your household own any irons (charcoal or electric)? (0 = no, 5 = yes) | 0.451 | 0.218 | 0.233 *** |
| | (0.498) | (0.414) | [0.027] |
| How many tables does your household own? (0 = none, 2 = one, 4 = two, 6 = three or more) | 3.007 | 2.392 | 0.628 *** |
| | (1.463) | (1.447) | [0.090] |
| PPI score | 45.40 | 40.51 | 5.002 *** |
| | (10.24) | (11.75) | [0.708] |
| Poverty likelihood (USD 1.9/person/day) | 31.81 | 40.84 | −9.264 *** |
| | (18.32) | (22.01) | [1.313] |

Standard deviation in parenthesis for columns 2–3. Standard errors in parenthesis for column 4. *** And * denote significance at 1% and 10%, respectively. MM stands for mobile money.

The study households' demographic, socioeconomic, and institutional characteristics constituted the independent variables identified based on a review of similar empirical impact studies carried out in the past (e.g., Ouma et al. 2017).

## 3. Results and Discussions

### 3.1. Descriptive Statistics

Table 1 presents the descriptive characteristics of the sample households. Most households were male-headed, comprising a household head who was middle-aged with a primary level of education. The average household size was five individuals. The family was the primary source of farm labor for different agricultural activities. Agricultural production was the main occupation of most household heads, who operated an average farm holding of 4.5 acres and livestock holdings of 1.2 Tropical Livestock Units (TLUs). Maize was the most important crop grown by the sampled households. On average, about 22% of households had access to a water source for irrigation. Household heads were also involved in social networks through participation in credit associations.

About 72% of household heads reported having used MM services in 2017. Most household characteristics were significantly different between MM users and non-users. MM users tended to be male heads of households, relatively younger, more formally educated, and wealthier, as measured by TLUs, with more social networks and access to credit and tractors. For example, 26% of MM users had access to credit compared to only

17% of non-users. Similarly, 43% of MM users had access to tractor services compared to 29% of non-users. According to Kim (2021), being younger motivates the use of MM services, while being wealthier and educated equips users with financial resources to buy handsets to deposit and access money and effectively engage with the different functions available on the MM platform (Amoah et al. 2020; Sekabira and Qaim 2017a). In addition, a more significant proportion of MM users had longer membership periods (months) in VICOBA and were wealthier than non-users, as measured by the value of their shareholdings. MM users also tended to be more networked within VICOBA (knew more than half of the group members) than non-users.

Regarding outcome variables, MM users tended to have a significantly higher average maize productivity and a lower average poverty likelihood than non-users without keeping other factors constant. There were also significant differences between MM users and non-users in the values of indicators used to construct the PPI scores (Table 2). MM users had significantly higher PPI scores and a lower poverty likelihood than non-users. Similar results have been established in SSA countries, including Burkina Faso (N'dri and Kakinaka 2020), Kenya (Suri and Jack 2016), and Uganda (Munyegera and Matsumoto 2018), and in another study in Tanzania (Abiona and Koppensteiner 2020). These lower poverty likelihoods might have been attained through the ability of MM users to save, reduce transactions costs, mitigate financial shocks, and engage in non-agricultural businesses, among other factors (see Abiona and Koppensteiner 2020; Suri and Jack 2016).

Given the heterogeneities in household characteristics between MM users and non-users (Tables 1 and 2), the differences in maize productivity and poverty likelihood between the two groups did not have a causal interpretation. In other words, the estimated maize productivity and poverty likelihood differences cannot be attributed to MM usage as the heterogeneities in the household characteristics might confound the effects of the MM usage. The differences could be due to the differences in household characteristics.

*3.2. ESR Model Diagnostics*

The model diagnostics results suggest that the hypothesis of no sample selectivity bias can be rejected, indicating the existence of self-selection bias in MM usage. The correlation coefficient between the error term of the selection equation (MM usage) and that of the maize productivity equation in the ESR framework was negative and statistically significant for MM users ($\rho_{e1u} = -1.33$; $p < 0.01$) but positive and statistically insignificant for non-users (Table A1 in Appendix A). The negative correlation sign for MM users in the maize productivity model indicates a positive selection bias, implying that farmers with above-average maize productivity are more likely to use MM services.

A similar analysis for non-users shows that the correlation between the error term of the selection equation and that of the poverty likelihood equation was positive and statistically significant ($\rho_{e2u} = 0.51$, $p < 0.05$) but negative and statistically insignificant for MM users (Table A2 in Appendix A). The positive correlation sign for MM non-users in the poverty likelihood model implies a negative selection bias, indicating that farmers with a lower-than-average probability of being poor are more likely to use MM services. The alternate signs of the correlation coefficients in the maize productivity and poverty likelihood models suggest that farmers self-select MM or alternative sources according to comparative advantage. This implies that MM usage would result in higher maize productivity and a lower poverty likelihood than that achieved under random assignment (Maddala 1983).

Two instruments that facilitate access to information (social capital and access to input suppliers) were used to identify the causal effects of MM usage. By definition, an instrument is directly related to the treatment variable (MM usage) but indirectly related to the outcome equation. Social capital, defined as the VICOBA social network, was used in the maize productivity model because a social network can directly correlate with MM usage but is unlikely to affect maize productivity directly. Farmers with an extensive network of fellow VICOBA members are more likely to use MM services since they quickly

access information. These farmers may not necessarily be more productive than those with no social network. Social networks have previously been used as instrumental variables and found to be positively associated with MM usage (Mukong and Nanziri 2021; Kiconco et al. 2020; Murendo et al. 2018).

Similarly, access to input suppliers, defined as proximity to farm input shops, was used in the poverty likelihood model. Access to input shops is likely to correlate with MM usage directly but is unlikely to affect poverty likelihood directly. Farmers with access to input shops are more likely to use MM in that, in most cases, input shops serve as local MM agents for cash depositing, sending, and withdrawing. MM agents are a strong determinant of MM usage (Hamdan et al. 2021; Kikulwe et al. 2014; Kirui et al. 2013). However, such farmers may not necessarily have a lower poverty likelihood than fellow farmers far away from input shops. The farmers near input shops may be expected to have a lower poverty likelihood only through access to the input shops. Proximity to input shops was used as an instrument by Munyegera and Matsumoto (2018) in studying the role of MM services in rural Uganda. Past studies used service providers as instrumental variables for casual impact identification (Shiferaw et al. 2014).

We conducted a falsification test on the validity of the instruments following Di Falco et al. (2011). The test results show that social capital directly influenced MM usage (chi-square = 13.62, *p* = 0.00) but did not affect the maize productivity outcome (F-stat = 1.84, *p* = 0.17). Similarly, the effect of access to input shops was statistically significant in the MM selection equation (chi-square = 11.68; *p* = 0.00) but not in the poverty likelihood equation (F-stat = 1.25; *p* = 0.26), indicating that the instruments are not statistically significant drivers of maize productivity and poverty likelihood. They may affect the outcome variables only through MM usage.

### 3.3. Determinants of MM Usage Based on the ESR Model

The parameter estimates of the determinants of the MM usage come from the first-stage (Equation (3)) estimation of the ESR model presented in the second columns of Tables A1 and A2 in Appendix A. Results show that MM usage was strongly and positively associated with the education level of the household head, television ownership, land holdings, tractor access, credit access, input access, and social networks in VICOBA. Consistent with expectations, the age of the household head was inversely correlated with MM usage, while the education of the household head was positively correlated with MM usage. This result is consistent with findings in many studies (Kumar and Pathak 2022; Sekabira and Qaim 2017a; Kikulwe et al. 2014; Kirui et al. 2013). The results also show a strong relationship between social networks and MM usage, which suggests that households with social networks in VICOBA can quickly receive information on MM services and new MM products and features (Okello et al. 2018; Kirui et al. 2013). In addition, the study does not show a statistically significant relationship between household type (male-headed versus female-headed) and MM usage. This result supports the findings by Chamboko (2022), who found that household type is not a significant determinant of using MM services.

### 3.4. Determinants of Maize Productivity and Poverty Likelihood Based on the ESR Model

The FIML parameter estimates of the determinants of maize productivity and poverty likelihood (Equations (3) and (4)) are presented in the third and fourth columns of Tables A1 and A2 in Appendix A. The third and fourth columns of Table A1 show the parameter estimates of the control variables for maize productivity for MM users and non-users, respectively. Similarly, the third and fourth columns of Table A2 present the parameter estimates of the control variables for poverty likelihood for MM users and non-users, respectively.

The ESR model of maize productivity results show noticeable differences in the signs, magnitudes, and statistical significances of the parameter estimates of some of the control variables, such as household type, education of the household head, and membership in VICOBA. Similarly, the ESR poverty likelihood model results show noticeable differences

in the signs, magnitudes, and statistical significances of the parameter estimates of other control variables, such as age and educational level of the household head, household type, access to credit, and social networks. These results mean that the variables have different returns depending on whether the household is an MM user or a non-user. For example, household type was significantly associated with maize productivity for non-users but not for users. Amoah et al. (2020) found a similar result in their study in Ghana. In contrast, the educational level of the household head was significantly associated with maize productivity for MM users but not for non-users. The literature has reported improved maize yields among mobile phone users (Issahaku et al. 2018).

Similar results were also observed in the poverty likelihood model. Some variables were significantly associated with poverty likelihood for MM users but not for non-users and vice versa. For example, access to credit was negatively and significantly related to poverty likelihood for MM users but not for non-users. MM users attain better financial resilience through savings, remittances, limited risks of economic shocks, and reduced transactions costs, thus reducing poverty (Hamdan et al. 2021; N'dri and Kakinaka 2020; Abiona and Koppensteiner 2020; Amoah et al. 2020; Boamah and Murshid 2019). In contrast, access to output buyers was positively and significantly associated with poverty likelihood for MM users but not for non-users. The differential returns to the same observed characteristics between MM users and non-users confirm the appropriateness of the switching regression framework (i.e., the existence of two regimes for the same outcome variable—one for MM users and another for non-users) used in the study. Given systematic differences between MM users and non-users, we applied the IV approach (ESR model) to address the unobserved heterogeneity and estimate the average treatment effects of the treatment (i.e., MM usage).

*3.5. Impacts of MM Usage on Maize Productivity and Poverty Likelihood*

The cumulative observed distribution of maize productivity for MM users lies predominantly to the right of the corresponding counterfactual distribution, while that of poverty likelihood lies to the left of the corresponding counterfactual distribution (Figures 2 and 3). This means that the cumulative distribution functions of maize productivity and poverty likelihood for MM users dominated the respective cumulative distribution functions for non-users at all maize productivity and poverty likelihood levels. This was confirmed by the Kolmogorov–Smirnov (K-S) test, which showed that the value for maize productivity and poverty likelihood were statistically significant, implying that the observed and counterfactual distributions of the outcome variables are not the same (Greene 2003). This result suggests that conditional on observables, MM users are likely to have higher maize productivity and a lower poverty likelihood on average than non-users.

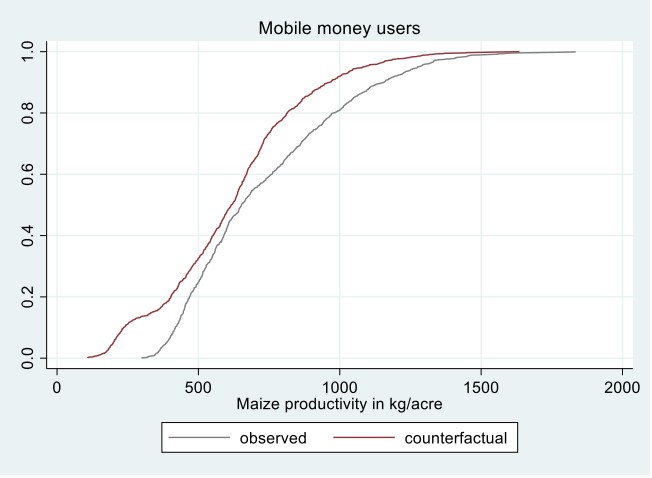

**Figure 2.** Observed and counterfactual cumulative distribution of maize productivity.

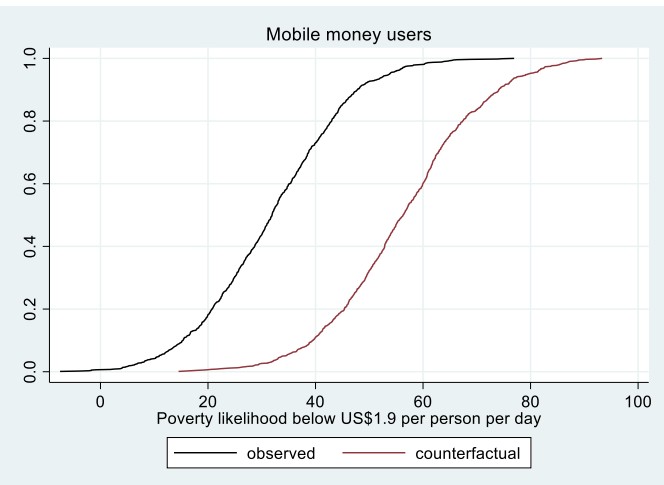

**Figure 3.** Observed and counterfactual cumulative distribution of poverty likelihood.

The average effects of MM usage (ATT) on maize productivity and household welfare are presented in Table 3. Results indicate that MM usage positively correlated with higher maize productivity and a lower poverty likelihood. The observed maize productivity with MM usage was about 680 kg/acre. In contrast, the counterfactual maize productivity was 556 kg/acre. This means that had MM users not used MM services, they would have produced 124 kg per acre lower maize yields, implying that MM usage led to a 22% maize productivity gain. This is consistent with Peprah et al. (2020), who found MM usage to have an enhancing effect on farm outputs in Ghana. The link between MM usage and maize productivity could be explained by the role that MM services play in facilitating secure saving, transferring, and receiving money, remittances, and payment, which is important in acquiring inputs for enhanced maize productivity.

**Table 3.** Average treatment effects of MM usage on maize productivity and poverty likelihood.

| Outcome | MM Usage | MM Non-Usage | ATT |
|---|---|---|---|
| Maize productivity (kg/acre) | 679.85 | 555.58 | 124.27 (9.94) *** |
| Poverty likelihood (%) | 31.94 | 56.54 | −24.59 (−39.87) *** |

Notes: absolute value of t-statistic in parenthesis: *** denote significance at 1%; MM stands for mobile money; the ATTs are presented in plain numbers, as calculated simply by taking the exponent of the maize productivity under observed and counterfactual conditions which are in natural logs in the model, and taking the mean difference between the two conditions.

Similarly, the poverty likelihood for MM users was nearly 32% compared to 57% for non-users. This means that MM usage led to a 25 percentage points reduction in the probability of falling below the poverty line at USD 1.9 per capita per day. This finding is consistent with Suri and Jack (2016), who found a positive relationship between MM usage and per capita consumption in Kenya, resulting in 2% of poor households being lifted out of poverty. This is also consistent with a finding in Bangladesh, where Hussain et al. (2019) found a positive correlation between financial account ownership and likelihood of financial resilience. They found that account holders' chances of being financially resilient were around 1.4 times higher than those of their counterparts. A similar study in rural China by Wang et al. (2022) revealed that MM usage significantly increased farmers' livelihoods, as measured by farm income and per capita consumption. Peprah et al. (2020) found a similar effect of MM usage in Ghana.

Moreover, MM usage was found to be positively associated with off-farm income accumulation in Uganda (Sekabira and Qaim 2017a). A plausible explanation for the better performance of MM users is a reduction in transaction costs associated with MM usage, savings, and productive investments in farm inputs (Munyegera and Matsumoto 2016, 2018; Sekabira and Qaim 2017a, 2017b; Suri and Jack 2016). The reduction in transaction

costs is associated with more household income availability, leading to a lower poverty likelihood (N'dri and Kakinaka 2020; Mwalupaso et al. 2019; Issahaku et al. 2018; Sekabira and Qaim 2017b).

*3.6. Impacts of MM Usage by Household Type*

The results in Table 4 indicate that MM usage has increased maize productivity and a reduced poverty likelihood in female-headed households (FHHs) and male-headed households (MHHs). Nonetheless, the ATT on maize productivity was nearly 116 kg/acre higher for FHHs than for MHHs, while the ATT on poverty likelihood was lower for FHHs than for MHHs by 9 percentage points. The differences in the ATT values for maize productivity and poverty likelihood between the two household types were statistically significant. This result is consistent with a similar study in Kenya, where Suri and Jack (2016) found that the increased use of MM services significantly helped reduce poverty in FHHs. Hussain et al. (2019) also found a significant correlation between gender and financial resilience, with men being 1.4 times more resilient than women. Furthermore, Kim (2021) showed that MM benefitted women, particularly those with lower educational attainment and income levels.

**Table 4.** Impacts of MM usage on maize productivity and poverty likelihood by household type.

| | MM Usage | MM Non-Usage | ATT | Mean Differences between FHHs and MHHs |
|---|---|---|---|---|
| | | Maize productivity (kg/acre) | | |
| FHH | 572.27 | 347.54 | 224.73 (7.71) *** | 115.95 (10.91) *** |
| MHH | 692.14 | 583.36 | 108.78 (8.30) *** | |
| | | Poverty likelihood (%) | | |
| FHH | 28.66 | 61.19 | −32.53 (15.79) *** | −8.75 (10.70) *** |
| MHH | 32.28 | 56.05 | −23.77 (36.94) *** | |

Notes: absolute value of t-test in parentheses; MM stands for mobile money; FHHs (female-headed households); MHHs (male-headed households); *** denote significance at 1%.

## 4. Conclusions and Implications

This study examined the determinants and impacts of MM usage on maize productivity and poverty likelihood by applying the ESR model to data collected from 1310 households in the Mbeya Region of Tanzania. Model diagnostic results suggest that the hypothesis of no sample selectivity bias can be rejected, indicating the existence of self-selection bias in MM usage, and confirming the appropriateness of the ESR model used in this study. If OLS estimation had been used instead, selection bias would have resulted in biased and inconsistent parameter estimates, leading to erroneous impact estimates.

The analysis of the determinants of MM usage showed a strong correlation between MM usage and the education of the household head, asset ownership, credit access, input access, and social networks in VICOBA. The results revealed gaps in MM usage between different household types based on the sex of the household head. Over 60% of FHHs in the sample used MM, compared to about 74% of MHHs. The gap in MM usage based on the sex of the household head calls for a targeted approach to reaching and supporting non-users in FHHs. The targeted approach of supporting rural households, particularly FHHs, who currently have relatively lower access to formal banking systems, befits the goal of financial inclusion.

The study also revealed differences in determinants of maize productivity between MM users and non-users. The education level of the household head, credit access, access to output buyers, and membership period strongly correlated with maize productivity for MM users. In contrast, the age of the household head, household type (male-headed), marital status, and group membership strongly correlated with maize productivity for non-users of MM.

Similarly, total land holdings, household size, and the dependency ratio strongly correlated with poverty likelihood for MM users and non-users. However, the age and education level of the head, education, credit access, access to output buyers, and social networks were strongly related to poverty likelihood for MM users. In contrast, household type and tractor access were significantly related to poverty likelihood for non-users.

The non-uniform effects of the determinants (besides MM usage) of maize productivity and poverty likelihood between MM users and non-users call for a differentiated approach instead of a one-size-fits-all approach. Considering the non-uniform effects of significant determinants between the two groups would contribute to the design of effective productivity improvement and poverty reduction programs. In terms of impacts, MM usage increased maize productivity and a reduced poverty likelihood. The results showed that MM usage led to a 22% gain in maize productivity and a 25-percentage point reduction in the probability of falling below the poverty line at USD 1.9 per capita per day. A comparison of impacts on maize productivity and poverty likelihood between MM users in the two household types showed that FHHs performed better than MHHs in both outcomes. FHHs obtained nearly 116 kg/acre more yield than MHH, resulting in a 9-percentage point difference in poverty likelihood between them. Together, these findings suggest that enhancing MM usage among households with constrained access to formal financial services can increase maize productivity and reduce poverty likelihood. The positive impacts of MM usage on increased maize productivity and a reduced poverty likelihood could increase the demand for mobile financial services, thus encouraging service providers to reach more rural villages. This could alleviate the lack of banking infrastructure in the rural areas of Tanzania in the long term. The reason for the differential effects of MM usage between FHHs and MHHs warrants further research.

**Author Contributions:** Conceptualization, H.K., S.F. and S.C.; methodology, H.K.; software, S.F.; validation, S.F., S.C., T.A., H.S. and V.M.; formal analysis, H.K.; investigation, S.F.; resources, S.F., S.C., T.A., H.S. and V.M.; data curation, H.K.; writing—original draft preparation, H.K.; writing—review and editing, S.F., S.C., T.A., H.S. and V.M.; visualization, H.K.; supervision, S.F. and S.C.; project administration, S.F., T.A. and V.M.; funding acquisition, S.F., T.A. and V.M. All authors have read and agreed to the published version of the manuscript.

**Funding:** This research received no external funding.

**Informed Consent Statement:** Informed consent was obtained from all subjects involved in the study.

**Data Availability Statement:** The data presented in this study are available on request from the corresponding author. The data are not publicly available due to organization policy.

**Acknowledgments:** Data used in this study was collected through a project titled "Improving Smallholder Tanzanian Farmers' Access to Improved Storage Technology and Credit (PTE Federal Award No. AID-OFDA-G-16-00261)." The project was a joint initiative of Purdue University and IITA. We appreciate the contribution of Jacob Ricker-Gilbert of Purdue University and Hira Chana of World Bank for leading the design of the survey tool employed to collect data used in this study. Special thanks to IITA and also Bernadette Majebelle and PHIRETAJO VICOBA (a local private NGO based in Mbeya Tanzania) for their support to the project.

**Conflicts of Interest:** The authors declare no conflict of interest.

## Appendix A

**Table A1.** FIML estimates of the ESR model of MM and maize productivity.

| Variables | MM Usage (Equation (3)) | Maize Productivity (kg/acre) | |
| --- | --- | --- | --- |
| | | MM Users (Equation (4a)) | Non-Users (Equation (4b)) |
| Age of the head | 0.018 | 0.010 | −0.028 * |
| | (0.018) | (0.013) | (0.016) |
| Age squared | −0.000 | −0.000 | 0.000 * |
| | (0.000) | (0.000) | (0.000) |
| Household type (1 = male-headed) | 0.148 | 0.024 | 0.524 *** |
| | (0.167) | (0.114) | (0.161) |
| Education level of the head (1 = primary education) | 0.582 *** | −0.261 *** | −0.005 |
| | (0.124) | (0.100) | (0.131) |
| Marital status of the head | 0.286 | −0.168 | −0.395 ** |
| | (0.187) | (0.134) | (0.179) |
| Household size | 0.004 | −0.010 | −0.002 |
| | (0.026) | (0.017) | (0.026) |
| Total land holdings (acres) | −0.027 *** | 0.005 | −0.003 |
| | (0.009) | (0.006) | (0.010) |
| TLU | 0.035 | 0.000 | 0.017 |
| | (0.024) | (0.012) | (0.027) |
| Television ownership (1 = yes) | 0.490 *** | 0.039 | 0.151 |
| | (0.112) | (0.062) | (0.170) |
| Credit access (1 = yes) | 0.223 ** | −0.102 * | −0.064 |
| | (0.094) | (0.059) | (0.108) |
| Extension access (1 = yes) | −0.123 | 0.079 | 0.057 |
| | (0.083) | (0.055) | (0.084) |
| Group membership (1 = yes) | 0.109 | 0.072 | 0.248 * |
| | (0.121) | (0.075) | (0.136) |
| Access to irrigation water (1 = yes) | −0.025 | 0.044 | 0.021 |
| | (0.091) | (0.060) | (0.093) |
| Access to tractor (1 = yes) | 0.207 ** | 0.014 | −0.110 |
| | (0.094) | (0.060) | (0.111) |
| Access to input shop (1 = yes) | −0.305 *** | −0.094 | −0.160 |
| | (0.097) | (0.066) | (0.106) |
| Access to output buyers (1 = yes) | −0.071 | 0.111 * | −0.086 |
| | (0.091) | (0.061) | (0.090) |
| Period of membership in VICOBA (months) | 0.016 | 0.025 * | 0.034 |
| | (0.021) | (0.013) | (0.023) |
| Wealth status (share proportion in VICOBA) | 0.222 | 0.148 | 0.292 |
| | (0.212) | (0.129) | (0.237) |
| Social networks in VICOBA | 0.216 *** | | |
| | (0.077) | | |
| Constant | −0.635 | 6.520 *** | 6.655 *** |
| | (0.453) | (0.321) | (0.437) |
| $\sigma_{e1}$ | | −0.213 *** | |
| | | (0.031) | |
| $\sigma_{e2}$ | | | −0.356 *** |
| | | | (0.037) |
| $\rho_{e1u}$ | | −1.330 *** | |
| | | (0.119) | |
| $\rho_{e2u}$ | | | 0.019 |
| | | | (0.342) |
| LR test of independent equations | | | 32.62 |
| Prob. > chi2 | | | (0.000) |
| Observations | 1310 | 1310 | 1310 |

Standard errors in parentheses; ***, **, and * denote significance at 1%, 5%, and 10%, respectively MM stands for mobile money.

**Table A2.** FIML estimates of the ESR model of MM and poverty likelihood.

| Variables | MM Usage (Equation (3)) | Poverty Likelihood USD 1.9 per Capita per Day | |
|---|---|---|---|
| | | MM Users (Equation (4a)) | Non-Users (Equation (4b)) |
| Age of the head | 0.026 | 0.584 ** | −0.518 |
| | (0.019) | (0.254) | (0.444) |
| Age square | −0.000 * | −0.007 *** | 0.003 |
| | (0.000) | (0.003) | (0.005) |
| Household type (1 = male-headed) | 0.131 | −1.335 | −13.755 *** |
| | (0.176) | (2.107) | (4.343) |
| Marital status of the head (1 = married) | 0.320 | −3.157 | 5.021 |
| | (0.195) | (2.522) | (4.591) |
| Education level of the head (1 = primary education) | 0.637 *** | −4.377 ** | −4.290 |
| | (0.125) | (2.050) | (3.186) |
| Total land holdings (acres) | −0.022 ** | −0.394 *** | −0.741 *** |
| | (0.010) | (0.118) | (0.255) |
| TLU | 0.030 | −0.043 | 0.485 |
| | (0.026) | (0.206) | (0.695) |
| Household size | −0.022 | 5.750 *** | 6.748 *** |
| | (0.028) | (0.324) | (0.694) |
| Dependency ratio | 0.014 | 8.962 *** | 5.387 *** |
| | (0.063) | (0.741) | (1.523) |
| Credit access (1 = yes) | 0.272 *** | −5.315 *** | −1.632 |
| | (0.098) | (1.079) | (2.803) |
| Tractor access (1 = yes) | 0.294 *** | −1.126 | 5.522 * |
| | (0.100) | (1.105) | (2.965) |
| Group membership (1 = yes) | 0.167 | −0.536 | −1.196 |
| | (0.126) | (1.338) | (3.565) |
| Access to irrigation water (1 = yes) | −0.029 | −0.872 | −2.370 |
| | (0.096) | (1.097) | (2.439) |
| Access to output buyers (1 = yes) | −0.012 | 3.010 *** | 0.950 |
| | (0.096) | (0.992) | (2.321) |
| Period of membership in VICOBA (months) | 0.022 | −0.334 | −0.332 |
| | (0.021) | (0.233) | (0.595) |
| Share proportions in VICOBA | 0.346 | −0.293 | −2.593 |
| | (0.220) | (2.328) | (6.168) |
| Social networks in VICOBA | 0.348 *** | −2.266 * | 3.758 |
| | (0.093) | (1.218) | (2.547) |
| Access to input shops (=yes) | −0.379 *** | | |
| | (0.097) | | |
| Constant | −0.751 | −37.672 *** | 18.217 |
| | (0.607) | (7.929) | (14.074) |
| $\sigma_{e1}$ | | 2.623 *** | |
| | | (0.024) | |
| $\sigma_{e2}$ | | | 2.966 *** |
| | | | (0.085) |
| $\rho_{e1u}$ | | −0.124 | |
| | | (0.143) | |
| $\rho_{e2u}$ | | | 0.508 ** |
| | | | (0.254) |
| LR test of independent equations. | | | 3.01 |
| Prob. > chi2 | | | (0.0826) |
| Observations | 1310 | 1310 | 1310 |

Standard errors in parentheses; ***, **, and * denote significance at 1%, 5%, and 10%, respectively; MM stands for mobile money.

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
