# Peer review of "Maize Productivity and Household Welfare Impacts of Mobile Money Usage in Tanzania"

_ijfs, doi:10.3390/ijfs11010027_

Round 1
Reviewer 1 Report
The authors present an interesting manuscript. The use of mobile money (MM) is considered an important mobile financial service that is currently contributing to the expansion of microbusinesses, especially in developing economies.
There are conclusions in the article that require more discussion. At the end of the manuscript (Conclusion) it is necessary to state what other factors (besides MM) can influence the productivity of maize cultivation and poverty likelihood. What new insights and what future directions does the authors' study suggest?
The introduction was generally well written and delivered the knowledge clearly to the readers.
Authors may need to add the hypotheses (research questions) in the Introduction. It is missing the explicit testing of hypotheses in the present manuscript.
The manuscript must provide a clear and concise synthesis of the data – how do the calculations and information presented in your manuscript contribute to the advancement of scientific knowledge of the topic?
From a statistical point of view, the manuscript is fine (technical data analysis). Unfortunately, the fundamental analysis of the data is insufficiently performed and discussed (the situation in the Mbeya region, a more detailed analysis of the clients of the community bank VICOBA). It is recommended to carry out a better fundamental analysis of the examined sample of VICOBA bank clients).
Reviewer 2 Report
Referee Report:
“Maize Productivity and Household Welfare Impacts of Mobile Money Usage in Tanzania”
General Comments:
This empirical study examines the relationship between the use of mobile money on the productivity of maize farmers in a rural region of Tanzania. The authors’ methodology is well-designed given the available data and logical in its approach to the issues under examination. Overall, the paper is also well-written and easy to follow and understand. The paper contributes to the growing literature on the impact of mobile money on access to financial services and economic markets for those low-income groups who otherwise may be unbanked. Given my positive impression of the paper, my specific comments are mostly limited to issues of style and presentation. These recommended edits are offered to improve the overall readability of the paper.
Specific Comments:
1. There are several instances where the authors use an abbreviation or acronym before defining it. The abbreviation or acronym should be presented immediately after first use. This is true for both the abstract and the text of paper (since they may be viewed by readers and researchers separately).
a. Line 3 (in the abstract); “MM usage” should be replaced with “mobile money (MM) usage”
b. Lines 28 and 31 (in the body of the text); note that the “(MM)” should be moved up from line 28 to 31 where it is first used in the body of the text
c. Line 98; “ESR model” should be replaced with “endogenous switching regression (ESR)” as this is the first use of the term in the body of the text
2. In the paragraph composing lines 133 to 142; the word “alternative” should in all cases be plural . . . “alternatives” as multiple alternatives to using mobile money exist – there is just not one alternative choice.
3. Lines 144 & 160; these should not be indented but started flush with left margin.
4. A blank line should be inserted above Line 210 and a blank line inserted below Line 211.
5. The top line of the box enclosing Figure 1 is missing. This should be fixed.
6. Table 1 only includes the means for each variable. It is standard practice to include below the means the Standard Deviations in parentheses. This is appropriate to help the reader truly understand the distribution of each variable. The mean does not provide a lot of information by itself. Furthermore, the table’s caption refers to “Descriptive statistics” . . . please add the standard deviations.
7. Line 300; “SSA” is not defined. I assume this is “Sub-Sahara Africa” . . . please confirm and add to this to the text as this again the first use of an abbreviation.
8. Section 3.6; the abbreviations FHH and MHH are not defined. I assume these are “Female Headed Household” and “Male Headed Household” . . . please confirm and add to these to the text on Line 471 as this again these are first use of abbreviations.
9. Throughout the paper, the tables are poorly placed and many are split over two pages. This makes it very difficult for the reader. Please ensure that the tables are properly placed and wherever possible are limited to appearing on a single page. Also, reformatting of the notes below the table look very odd and should be aligned with the left margin of the table, not aligned with the text that follows the table.

Author Response
Please see the attachnent
